# Short-Term Effect of SARS-CoV-2 Spike Protein Receptor-Binding Domain-Specific Antibody Induction on Neutrophil-Mediated Immune Response in Mice

**DOI:** 10.3390/ijms23158234

**Published:** 2022-07-26

**Authors:** Elena L. Bolkhovitina, Julia D. Vavilova, Andrey O. Bogorodskiy, Yuliya A. Zagryadskaya, Ivan S. Okhrimenko, Alexander M. Sapozhnikov, Valentin I. Borshchevskiy, Marina A. Shevchenko

**Affiliations:** 1Department of Immunology, Shemyakin and Ovchinnikov Institute of Bioorganic Chemistry, Russian Academy of Sciences, 117997 Moscow, Russia; alenkash83@gmail.com (E.L.B.); juliateterina12@gmail.com (J.D.V.); amsap@mail.ru (A.M.S.); 2Research Center for Molecular Mechanisms of Aging and Age-Related Diseases, Moscow Institute of Physics and Technology, 141700 Dolgoprudny, Russia; bogorodskiy173@gmail.com (A.O.B.); 1989july@mail.ru (Y.A.Z.); ivan.okhrimenko@phystech.edu (I.S.O.); borshchevskiy.vi@phystech.edu (V.I.B.)

**Keywords:** SARS-CoV-2 infection, mouse model, receptor-binding domain, antibodies, neutrophils

## Abstract

Vaccination protects against COVID-19 via the spike protein receptor-binding domain (RBD)-specific antibody formation, but it also affects the innate immunity. The effects of specific antibody induction on neutrophils that can cause severe respiratory inflammation are important, though not completely investigated. In the present study, using a mouse model mimicking SARS-CoV-2 virus particle inhalation, we investigated neutrophil phenotype and activity alterations in the presence of RBD-specific antibodies. Mice were immunized with RBD and a week after a strong antibody response establishment received 100 nm particles in the RBD solution. Control mice received injections of a phosphate buffer instead of RBD. We show that the application of 100 nm particles in the RBD solution elevates neutrophil recruitment to the blood and the airways of RBD-immunized mice rather than in control mice. Analysis of bone marrow cells of mice with induced RBD-specific antibodies revealed the increased population of CXCR2^+^CD101^+^ neutrophils. These neutrophils did not demonstrate an enhanced ability of neutrophil extracellular traps (NETs) formation compared to the neutrophils from control mice. Thus, the induction of RBD-specific antibodies stimulates the activation of mature neutrophils that react to RBD-coated particles without triggering excessive inflammation.

## 1. Introduction

Vaccination is one of the most effective approaches to fight a pandemic such as COVID-19. Classical protection mechanisms implicate pathogen-specific T-cell activation and antibody production. Thus, the correlation between the virus-specific CD4^+^ cell activation, SARS-CoV-2 spike protein receptor-binding domain (RBD)-specific IgG titers, and virus neutralization antibody titers together with increased IFN-γ production by activated CD8^+^ cells proved to be the key features of the mRNA (BNT162b, BNT162b) vaccine-induced protective immune response [1,2]. The induction of RBD-recognizing and neutralizing antibodies and IFN-γ production were tested to confirm the efficiency of adenovirus-based vaccines [3]. Moreover, memory B-cells producing RBD-specific neutralizing antibodies contribute to effective infection protection over a long time [4]. Again, the induction of RBD-specific antibodies is an essential requirement for inactivated SARS-CoV-2 vaccines [5]. Furthermore, passive transfer of RBD-specific antibodies prevented body weight loss and mortality in mice with SARS-CoV-2-induced inflammation [6]. The cases of successful convalescent serum therapy also support the importance of specific antibody induction in protection against SARS-CoV-2-induced complications [7].

Besides adaptive immune response modulation, vaccination also alters innate immune response to virus determinants. The concept of trained immunity implies the epigenetic and metabolic reprogramming of innate immune cell progenitors after vaccination that leads to innate immune memory and alters the innate immune cell functions during the response to pathogen [8]. Opposite to memory T- and B-cells, the innate immune memory is not antigen epitope-specific and therefore should have potential against a broad spectrum of pathogens. However, to date, the phenomenon has not been properly investigated [9,10]. Recently, using peripheral blood mononuclear cells (PBMC) from BNT162b vaccinated and non-vaccinated donors, the vaccination-associated decreased production of TNF-α, IL-1β, and IL-1 receptor antagonists in response to cell stimulation with SARS-CoV-2 or several TLR ligands was demonstrated [2]. The finding supports the evidence that vaccination alters monocyte functions, and these alterations can potentially reduce the severity of SARS-CoV-2-induced inflammation in the case of infection.

The demonstrated induction of trained immunity by mRNA vaccine against SARS-CoV-2 is novel, while the reprogramming of innate immune progenitors by live-attenuated vaccines is well characterized. Thus, long-term effects of BCG vaccination on blood neutrophil phenotype, such as increased activation marker expression and enhanced antimicrobial function, were reported for at least three months after vaccination [11]. Such potential of neutrophil reprogramming is of great interest in the context of COVID-19 because of the known pathogenic effect of neutrophils and neutrophil extracellular traps (NETs) during the SARS-CoV-2-induced inflammation [12,13,14]. Neutrophils can mediate the vaccination- and specific antibody-associated side effects such as thrombosis [15]. At the same time, vaccination and RBD-specific antibodies prevent the COVID-19-associated severe inflammation and complications, some of which are mediated by neutrophils [16]. Thus, the effects of SARS-CoV-2 antigen-specific antibody induction on neutrophils are an important but poorly investigated.

The present study compares the neutrophil-mediated immune response to inhaled model particles mimicking the physical characteristics of SARS-CoV-2 (the size of 100 nm) dissolved in the RBD solution in mice with and without induced RBD-specific antibodies. We estimate the influence of specific antibody induction on periphery blood, lung tissue, bronchoalveolar lavage (BAL), and bone marrow neutrophils, their phenotype and functional activity. Our data support the evidence of interplay between specific activation of adaptive and neutrophil-mediated innate immune response. The data can be useful for the explanation of both the protective and side effects of vaccination.

## 2. Results

### 2.1. Induction and Characterization of RBD-Specific Antibody Response

To induce RBD-specific antibodies, mice received three intraperitoneal (i.p.) injections of RBD firstly in a concentration of 15 µg/mouse/injection followed by another three i.p. injections two weeks later in a concentration of 50 µg/mouse/injection (Figure 1A). A slight but significant increase in RBD-specific IgM, but not IgG antibodies in the RBD-injected mice was detected two weeks after the last 15 µg RBD injection (Figure 1B,C, 3 × 15 µg); and a significant increase in both IgM and IgG was detected a week after the last 50 µg RBD injection (Figure 1B,C, 3 × 15 µg + 3 × 50 µg). At this time point, the B to T cell ratio was increased in mice that received RBD injections compared to the intact mice (Figure 1D,E). IgG antibodies were mostly represented by IgG1, while the level of IgG2a was 10-fold lower than that of IgG1, although significantly increased compared to the intact mice (Figure 1F,G). The level of serum IgA was mild, but significantly higher than that of the intact mice (Figure 1H). IgG2b and IgG3 were also detected in the sera of mice after the RBD injections (Appendix A). No significant difference in the IgE secretion was found between the RBD- and PBS-injected mice, and the intact mice (Appendix A).

Thus, using immunization with RBD, we induced elevation of RBD-specific antibodies. The observed IgG1 domination indicated Th2-mediated response activation [17].

### 2.2. Elevated Neutrophil Recruitment to the Bloodstream in Response to the Model Virus Particle Application in Mice with Induced RBD-Specific Antibodies

To estimate the influence of specific antibody induction on the neutrophil-mediated response, mice with RBD-specific antibodies and the control mice received via oropharyngeal (o.ph.) application that mimics the SARS-CoV-2 virus particles inhalation (Figure 2A). The portion of myeloid cells in blood and the portion of neutrophils from myeloid cells were identified in the peripheral blood 2 and 24 h after particle application according to the strategy developed by Liu et al. [18]. The modification of the strategy allowed the peripheral blood neutrophil portion identification (Figure 2B and Appendix A). Both the myeloid cell and the neutrophil from myeloid cell portions were elevated 2 h after the application of 100 nm particles in the RBD solution compared to the intact mice (Figure 2C,D). In mice with RBD-specific antibodies, the portion of neutrophils in the peripheral blood was also elevated, confirming the neutrophil-mediated response activation (Figure 2E). No elevation of myeloid cell, neutrophil from myeloid and neutrophil from peripheral blood portions was observed in the control mice compared to the intact mice (Figure 2C–E). RBD-injections without the application of 100 nm particles in the RBD solution did not have a prolonged effect on the neutrophil peripheral blood portion (Appendix A).

The level of angiotensin II was elevated in mice with induced RBD-specific antibodies and the control mice 2 h after the application of 100 nm particles in the RBD solution compared to the intact mice (Figure 2F). Interestingly, the level of angiotensin II in the periphery blood of mice with induced RBD-specific antibodies was significantly lower than that of the control mice (Figure 2F).

In mice with RBD-specific antibodies, the elevation of the neutrophil recruitment to the peripheral blood was observed in response to the application of 100 nm particles in the RBD-solution. These mice demonstrated lower levels of the peripheral blood serum angiotensin II that did not affect the neutrophil recruitment from the bone marrow to the blood. However, it could affect recruitment to the site of inflammation or functional activity.

### 2.3. Neutrophil-Mediated Immune Response to Inhaled Model Virus Particles in the Airways of Mice with Induced RBD-Specific Antibodies

To study the neutrophil-mediated immune response in the airways after the application of model particles, the immunohistochemistry of whole-mount lungs with subsequent imaging using fluorescent confocal laser scanning microscopy was performed. As we used fluorescent 100 nm particles (FluoSpheres), we were able to trace their distribution in the lungs. Staining of biotin-rich airway epithelial cells with fluorescently labeled phalloidin permitted us to visualize airways and staining against the adhesion molecule VCAM-1 (CD106) permitted the visualization of large vessels (Figure 3A). Imaging with higher magnification permitted us to locate CD11b cells in the perivascular space and inside the lung vessels of mice with induced RBD-specific antibodies (Figure 3B,C).

To check the presence of neutrophils in the bronchi, immunohistochemistry and imaging of the whole-mount conducting airway were performed. To visualize neutrophils, we used anti-Ly6G antibodies; using transgenic CD11c-EYFP mice permitted visualizing CD11c^+^ cells in the airways. We observed that FluoSpheres were mostly located in the luminal side of the airway epithelium and were internalized by the actin-rich cells, some of which were also CD11c^+^ (Figure 3D,E). Neutrophils were observed in the conducting airway walls of mice with induced RBD-specific antibodies (Figure 3D) and in the control mice (Figure 3E). The number of Ly6G^+^ cells in the conducting airway wall and the neutrophil percentage in BAL were significantly increased in mice with induced RBD-specific antibodies, but not in the control mice 24 h after the application of 100 nm particles in the RBD solution compared to the intact mice (Figure 3F,G).

The blood RBD-specific IgA level was significantly increased in mice with induced RBD-specific antibodies (Figure 1H). However, we did not find significant elevation of RBD-specific IgA in BAL fluids of these mice compared to the intact mice (Figure 3H). At the same time, the level of BAL fluid RBD-specific IgG in these mice was significantly elevated 24 h after the application of 100 nm particles in the RBD solution compared to the intact animals (Figure 3I).

Neutrophils recruited to the lung vessels and conducting airway walls of mice with induced RBD-specific antibodies in response to the application of 100 nm particles in the RBD solution. The BAL fluid RBD-specific antibodies in these mice are represented with IgG, but not IgA, and therefore IgG should opsonize the RBD-coated 100 nm particles after the inhalation.

### 2.4. Bone Marrow Neutrophil Activation and Maturation in Mice with Induced RBD-Specific Antibodies

To check the alteration of the activation and maturation of neutrophils in mice with induced RBD-specific antibodies, we investigated bone marrow neutrophil phenotype. The neutrophils were identified among bone marrow cells using the strategy described by Khoyratty et al. [19]. CXCR2 was used as a marker of the neutrophil readiness to migrate to the bloodstream from the bone marrow, and CD101 was used as a neutrophil maturation marker [19]. Thus, we firstly identified the portion of CXCR2^+^ circulating neutrophils and then checked the maturation status of these neutrophils (Figure 4A).

No significant differences in the portions of neutrophils among the bone marrow cells and the portions of CXCR2^+^ neutrophils were detected 24 h after the application of 100 nm particles in the RBD solution to mice with induced RBD-specific antibodies and the control mice compared to the intact mice (Figure 4B). However, the portion of mature neutrophils among the circulating neutrophils was significantly higher in mice with RBD-specific antibodies compared to both intact and control mice (Figure 4D).

Thus, 24 h after the RBD-coated 100 nm particle application, the portion of mature neutrophils among the circulating neutrophils is elevated in mice with induced RBD-specific antibodies.

### 2.5. The Reactivity of Bone Marrow Neutrophils from Mice with RBD-Specific Antibodies

NETs are known to trigger the uncontrolled immune response during COVID-19 [14]. To check the NET formation activity, 24 h after the application of 100 nm particles in the RBD solution, neutrophils were isolated from the bone marrow of the RBD-injected mice, the control mice, and the intact mice by the negative selection. Then, the neutrophils were activated with an ionophore (A23187) within 3 h. Extracellular nucleotides were detected using SytoxGreen staining [20]. Confocal imaging of the neutrophils 3 h after the incubation with A23187 revealed extracellular DNA extensions that were not observed in the primary culture of neutrophils in the absence of A23187 (Figure 5A,B). The quantitative analysis revealed a significantly lower level of extracellular nucleotides in the primary culture of neutrophils isolated from mice with induced RBD-specific antibodies compared to the control mice (Figure 5C). We performed staining for citrullinated histone H3, which confirmed that the observed DNA extensions were NETs (Figure 5 D–F).

The demonstrated results provide the evidence that specific antibody induction makes neutrophils more resistant to NETs formation.

## 3. Discussion

In the present study, we investigated the influence of specific to pathogenic determinant antibody induction on the neutrophil-mediated immune response. RBD injections were performed for the induction of specific antibodies and the o.ph. application of 100-nm particles in the RBD solution mimicked the virus loading. Using this approach, we demonstrated the enhanced neutrophil recruitment to the bloodstream and further to the airways in response to 100-nm particles with RBD in mice with (but not without) RBD-specific antibodies. Neutrophils are essential for bacterial and fungal killing, however the role of neutrophils in the antiviral response is still unclear [21,22,23]. In the respiratory tract of virus-infected patients, including patients suffering from severe COVID-19, intensive neutrophil infiltration was reported and thought to cause structural tissue damage and disease progression [12,13,14]. In the present study, the initial stage of infection was modeled and the neutrophil recruitment to the respiratory tract was less intensive than that of severe inflammation. Nevertheless, the observed increase of the neutrophil recruitment in the RBD-injected mice was unexpected and required further characterization of the neutrophils-mediated immune response in these mice.

Neutrophils recruit to the bloodstream and periphery from the bone marrow after the activation of the signaling through CXCR2; therefore, a high level of CXCR2 expression is detected on circulating neutrophils [24,25]. In the present study, 24 h after the application of 100 nm particles in the RBD solution, we identified bone marrow neutrophils with high expression of CXCR2 in mice with RBD-specific antibodies and in the control mice. The portions of circulating neutrophils did not significantly differ between the groups. However, the maturation state of the circulating neutrophils that was detected by CD101 expression [19] distinguished mice with RBD-specific antibodies from the control mice. As we reported here, 24 h after the application of the 100-nm particle in the RBD solution, more mature neutrophils were detected among the circulating neutrophils of the RBD-injected mice compared to the control animals.

As NETs mediate the pathology of COVID-19 [12,13,14], we tested the ability of neutrophils from mice with RBD-specific antibodies to form NETs. To compare the reactivity of neutrophils from the RBD-injected mice, the control mice and the intact mice, we used the calcium ionophore A23187, which is a known NET activator [20]. Net formation that was confirmed by anti-citrullinated histone H3 staining was observed in the primary culture of both the RBD-injected mice and the control mice and the intact mice. However, the quantitative analysis of the extracellular nucleotides indicated less extensity of netosis in the RBD-injected mice compared to the control mice 24 h after the application of 100-nm particles in the RBD solution.

Thus, our data demonstrate that neutrophils from mice with induced RBD-specific antibodies extensively egress from the bone marrow into circulation. Moreover, these neutrophils are more mature and demonstrate lesser reactivity. The observed phenomenon can be explained by the effect of antibodies on the neutrophil activity through the interaction with Fcγ receptors [26]. IgG1 that was abundant after the RBD injections can bind with higher affinity with the inhibitory FcγRIIB [27]. Although the FcγRIIB on neutrophils is low, it can alter upon vaccination due to the increased production of IFNγ that can induce the elevation of the FcγRIIB expression [1,2,3,26,28]. Taking this into account, we can suggest that the interaction of IgG-opsonized virus particles with FcγRIIB, which expression was enhanced by vaccination, could prevent neutrophil activation and subsequent NET formation that we observed in the present study. Additionally, IgG-opsonized virus particles can activate the classical complement pathway [29]. Both the lectin and classical pathways of complement activation play pathogenic role during COVID-19 and can induce platelet activation, netosis, and thrombosis [30,31,32]. However, the complement activation can also promote virus phagocytosis via CR3 (CD11b/CD18) or CR4 (CD11c/CD18) [33,34]. As we have shown here, monocytes that internalized 100 nm particles in conducting airway mucosa are CD11c-expressing cells, while recruited neutrophils CD11b-expressing; and they might participate in complement-mediated virus particle internalization.

Our data also demonstrated the decreased level of peripheral blood serum angiotensin II in the RBD-injected mice compared to the control mice 2 h after the application of 100 nm particles in the RBD solution. Angiotensin II contributes to the neutrophil recruitment and can trigger the NET formation [35,36]. In the present study, despite the lower angiotensin II level, we observed an increased neutrophil percentage in the blood of the RBD-injected mice 2 h after the application of 100 nm particle in the RBD solution. However, neutrophils from the bone marrow of the RBD-injected mice were less prone to netosis and the lower level of serum angiotensin II at the earlier stage of inflammation can be another explanation of decreased reactivity of neutrophils from the RBD-injected mice.

In accordance with the trained immunity concept, vaccination can induce the alteration of innate immune cell functions [8]. In the present study, we demonstrated the alteration of neutrophil functional activity, i.e., NET formation ability, during the specific antibody induction. Although our data suggest the antibody-mediated effects on neutrophil maturation and the suppression of neutrophil reactivity as a result of vaccination, it is worth mentioning that not all the antibodies have a protective effect in the inflammation. Autoantibodies, such as antinuclear antibodies (ANA) and antiplatelet autoantibodies (APA), are thought to be the causes of COVID-19 complications [37,38]. While APA are associated with thrombocytopenia, ANA can be responsible for the vasculitis in COVID-19 complications [39,40]. Interestingly, anti-neutrophil cytoplasmic antibodies (ANCA), directed against proteins of neutrophil cytoplasmic granules, were detected in COVID-19 patients [41,42]. ANCA can be induced by the NET remnants. Moreover, ANCA can facilitate netosis [43]. Several cases reports support the evidence of post-COVID-19 ANCA-associated vasculitis [44]. The data provided in the present study demonstrate that RBD-specific antibody induction suppresses NET formation, supporting the preventive effects of vaccination in the case of vascular complications. Unfortunately, a number of case reports demonstrate that neutrophil-associated complications and vasculitis can be triggered by vaccination [45,46]. Enhanced pathogen-induced neutrophil recruitment in the case of RBD-specific antibody induction observed in the present study can in part explain the adverse reactions to vaccination.

Thus, the effects of SARS-CoV-2 spike protein RBD-specific antibody induction are complex and not restricted to adaptive immunity activation. Among the short-term effects of specific antibody induction on innate immune response are the enhancement of neutrophil recruitment, circulating neutrophil maturation, and reactivity suppression.

## 4. Materials and Methods

### 4.1. Animals and Ethics Statement

F1 hybrid mice were produced by crossing CD11c-EYFP-C57BL/6 mice [47] kindly gifted to us by Prof. Michel C. Nussenzweig (The Rockefeller University, New York, NY, USA) and Prof. Armin Braun (Fraunhofer Institute for Toxicology and Experimental Medicine ITEM, Hannover, Germany) and BALB/c mice that were purchased from Animal Facility Stezar (Vladimir, Russian Federation). The mice were bred and housed in the vivarium of the Shemyakin and Ovchinnikov Institute of Bioorganic Chemistry, the Russian Academy of Sciences. In the study, females (18–25 g, 20–30 weeks) were used. All animal experiments were performed in concordance with the Guide for the Care and Use of Laboratory Animals under a protocol approved by the Institutional Animal Care and Use Committee at the Shemyakin and Ovchinnikov Institute of Bioorganic Chemistry, the Russian Academy of Sciences (protocol numbers 302/2020, 319/2021). The animals were given standard food and tap water ad libitum and housed under regular 12 h dark:light cycles at 22 °C.

### 4.2. RBD-Specific Antibody Induction

The mice received three i.p. injections of RBD (Hytest, Russia, 8COV1) in a concentration of 15 µg/mouse/injection in a volume of 200 µL DPBS (PanEco, Moscow, Russia) daily. For the subsequent two weeks, the mice received three i.p. injections of RBD in a concentration 50 µg/mouse/injection in a volume of 200 µL DPBS daily.

### 4.3. 100 nm Fluorescent Particles and RBD Application

Fluorescent particles FluoSpheres Carboxylate-Modified Microspheres, 100 nm, red fluorescent (580/605) (ThermoFisher, F8801, Waltham, MA, USA) were used in the study. The particle concentration was estimated in accordance with the manufacturer recommendations; the concentration of 1.7 × 10^8^ was used for the application to the mice. The particles were sonicated for 1 min, suspended in 0.1% RBD solution, and applied to the mice within 30 min. Mice were anesthetized by inhalation of 0.5–3% isoflurane (Karizoo, Barcelona, Spain), and a 50 μL droplet containing particles was applied to the oropharyngeal cavity of each mouse.

### 4.4. Blood Collection

The peripheral blood was collected from the tail vein two weeks after the last 15-µg RBD injection, one week after the last 50-µg injection and 2 h after the application of 100 nm particles in the RBD solution. Twenty-four hours after the application of 100 nm particles in the RBD solution, blood was collected from the vena cava of the euthanized mice. For the serum analyses, blood was collected to the 1.5 mL test-tubes, stored for 30 min at RT, then centrifuged at c.a. 500 g using CV 1500 (Biosan, Riga, Latvia) for 15 min. The supernatants were aliquoted and stored at −20 °C until use. For the blood cell cytometry analysis, blood was collected to the 1.5 mL test-tubes with 50 µL heparin (VelPharm, Moscow, Russia) and transferred to the 10 mL of the hemolysis buffer (155 mM NH_4_Cl (Reachem, Moscow, Russia); 0.1 mM Na_2_EDTA (Sigma-Aldrich, St. Louis, MO, USA); 10 mM NaHCO_3_ (PanReac Applichem); pH 7.3) stored at +4 °C and warmed to RT before use in 50 mL tubes. After 5 min at RT, 20 mL of DPBS was added, and the tubes were centrifuged at 380 g 5 min. The supernatants were replaced with 5 mL of the hemolysis buffer. After 5 min at RT, the cells were centrifuged at 380 g 5 min and transferred to 500 µL of the cytometry buffer (1% BSA, 2 mM EDTA). Then, the cells were centrifuged at c.a. 500 g for 10 min. The cell pellets were dissolved in 30 µL of the cytometry buffer and placed in the 96 well plate.

### 4.5. Blood Cell Analysis by Flow Cytometry

For the flow cytometry analysis of the peripheral blood cell detection, we used the strategy recommended by Liu et al. [18]. The samples were preincubated with anti-mouse CD16/CD32 (Miltenyi Biotec, Bergisch Gladbach, Germany, 130-092-574) for 15 min. Then the following antibodies (all from Miltenyi Biotec) were used: anti-mouse Ly6G–VioBlue (130-119-902), anti-mouse FcεR1–PE (130-118-896), anti-mouse SiglecF–PE-Vio615 (130-112-330), anti-mouse CD172–PE-Vio770 (130-123-154), anti-mouse CD45–APC (130-110-798), anti-mouse CD11b–APC-Vio770 (130-113-803). The antibodies were used in dilution 1:30. The samples were incubated for 30 min and washed twice with DPBS. SytoxGreen (Invitrogen, Eugene, OR, USA, S34859) was added (in dilution 1:1,000,000) 5 min before the acquisition. The measurement was performed using a MACSQuant Analyzer 10 (Miltenyi Biotec).

### 4.6. Bone Marrow Cells Analysis

The cells were washed from the tibia and femur of the mice using DPBS and centrifuged at 380 g 10 min. For neutrophil isolation, the cells were transferred to the buffer (1% BSA 2 mM EDTA). The rest of the cells were treated with a hemolysis buffer (3 min at RT) and washed twice with DPBS and transferred to the cytometry buffer. The cells were then incubated with the following antibodies (all from Myltenyi Biotec): anti-mouse Ly6G–VioBlue (130-119-902), anti-mouse SiglecF–PE-Vio615 (130-112-330), CXCR2–PE, and CD101–PE-Vio770. To exclude the precursors and the cells rather than granulocytes, the following antibodies were used and termed as Lineage (Lin): anti-mouse CD3–APC (130-122-943), anti-mouse CD19–APC (130-123-791), anti-mouse CD11c–APC (130-119-802), anti-mouse NK1.1–APC (130-117-349), anti-mouse Ter119–APC (130-102-290). Then, 30 min after, the cells were washed twice with DPBS, dissolved in SytoxGreen, and subjected to flow cytometry analysis.

### 4.7. Neutrophil Isolation

Neutrophils were isolated from the bone marrow cell suspension by the negative selection using magnetic separation. The following reagents and equipment were used (all from Miltenyi Biotec, Germany): Neutrophil Isolation Kit (mouse), separator QuadroMACS, and LD Columns. The separation was performed in accordance with the manufacturer recommendations.

### 4.8. Extracellular Nucleotide Detection and Imaging

The extracellular nucleotides were detected using SytoxGreen staining [20] Briefly, isolated from bone marrow cells neutrophils were transferred to the SytoxGreen dissolved in HBSS (PanEco, Moscow, Russia) 1:1000. Neutrophils were incubated in a black 96-well plate (SPL Life Sciences, Pocheon, Korea) in a concentration of 2 × 10^5^ cells per well at 37 °C and 5% CO_2_ in the presence of 5 µM of A23187 for 3 h. The samples were analyzed using GloMax Microplate reader Multisystem (Promega, Madison, WI, USA) in a fluorescence mode, filter 490 nm.

Neutrophils were also loaded to the coverglass-bottom slides (Ibidi, Gräfelfing, Germany), incubated 3 h in presence or without of 5 µM of A23187, then fixed with 4% paraformaldehyde (Applichem, Darmstadt, Germany). The samples were stained with anti-histone H3 (citrulline R2 + R8 + R17) antibody (Abcam, Boston, MA, USA, ab5103) and the secondary goat-anti-rabbit PE-conjugated (Sigma-Aldrich, St. Louis, MO, USA, P9537). The nuclei were stained with Hoechst (PanEco, Moscow, Russia), dilution 1:1000.

### 4.9. Bronchoalveolar Lavage (BAL) Collection

The animals were euthanized, and their lungs were washed with 800 µL of DPBS twice. The bronchus of the left lung lobe was tied with a surgical thread to prevent inflation with DPBS. BAL was collected to the 1.5 mL test-tube, centrifuged at c.a. 500 g for 10 min. The supernatants—the BAL fluids—were aliquoted and stored at −20 °C until use. The pellets were suspended in 500 µL and loaded onto the glass slides using centrifuge Cytospin 2 (Shendon, London, UK). Differential cell staining was performed using Diachem DiffQuick reagents (Abris, St. Petersburg, Russia) in accordance with the manufacturer recommendations.

### 4.10. Antibody Detection

The peripheral blood serum samples were tested for the RBD-specific IgG, IgG1, IgG2a, IgG2b, and IgG3 and the BAL fluid samples were tested for the RBD-specific IgG and IgA using ELISA. The ELISA plates were incubated with 5 µg/mL RBD dissolved in DPBS overnight at +4 °C. Washing with 0.5% Tween20-DPBS for three times was performed between each step of incubation. The plates were incubated with 1% BSA blocking buffer for 30 min, then with serum samples for 1 h, and then with respective HRP-conjugated antibodies for 1 h at RT. The TMB solution (Immunotek, Moscow, Russia) was used for visualization. Total IgE in the serum samples was detected using ELISA MAX Standard Set Mouse IgE (BioLegend, San Diego, CA, USA) in accordance with the manufacturer recommendations. The sample analysis was performed using Multiskan FC (Thermo Scientific, Waltham, MA, USA) at wavelengths of 450/690 nm.

### 4.11. Whole-Mount Lung Immunohistochemistry and Optical Clearing

Whole-mount lung staining and optical clearing was performed using Ce3D Tissue Clearing Kit (BioLegend, 427702) in accordance with the manufacturer recommendations. The following antibodies and reagents were used: anti-mouse CD11b-PE (BioLegend, 101208) in dilution 1:100, anti-mouse CD106 (BioLegend, 105712) in dilution 1:100, and streptavidin-Atto425 (Sigma-Aldrich, 09260) in dilution 1:200.

### 4.12. Whole-Mount Conducting Airway Specimen Preparation and Staining

The animals were euthanized, and their lungs were fixed with 2% paraformaldehyde without inflation and stored at +4 °C overnight. The main bronchi from the lung lobes were dissected. The left and right inferior lobes were then used for the quantitative analysis, the left middle or post-caval lobes were used for isotype-control staining. The airway specimens were then washed with DPBS, permeabilized with 0.3% Triton X-100, and blocked with 1% BSA. The following antibodies and dilutions were used: anti-mouse Ly6G-APC (BioLegend, 127614, dilution 1:50). For isotype-control staining APC-conjugated Rat IgG2a, κ (BioLegend, 400511), the specimens were incubated with antibodies overnight then washed with 0.3% Triton X-100 in DPBS and stained against actin by incubation for 1 h with Phalloidin-Atto425 (Sigma-Aldrich, 66939, dilution 1:50). If indicated, the nuclei were stained with Hoechst (dilution 1:1000) and subsequent washing with DPBS. All samples were covered with Prolong Gold mounting medium (Thermo Fisher, P36930).

### 4.13. Confocal Laser Scanning Microscopy (CLSM)

An inverted confocal LSM780 microscope (Zeiss, Jena, Germany) was used in all experiments with either a 10× (NA = 0.3), 40× (NA = 1.4, water immersive), or a 100× (NA = 1.46, oil immersive) objective. Excitation at 405, 488, 561, and 633 nm was used to visualize Atto425, EYFP, FluoSperes, and APC, respectively. The emission was measured in a CLSM λ-mode using a 34-channel QUASAR detector (Zeiss) set to a 405–695 nm range. For the quantitative analysis, the images were captured as 2 × 2 tile grids at the same regions of each specimen using the 40× objective, with an individual XYZ tile size of 354 µm × 354 µm × 30 µm. Higher magnification images were acquired in z-stacks at the region of interest using the 100× objective. The images were acquired as three-dimensional stacks. Spectral unmixing was performed using ZEN 2012 SP5 software (Zeiss). The images were created using Imaris version 9.8 software (Oxford Instruments, Zurich, Switzerland). Cells and FluoSpheres were presented either as maximum intensity projections or as surfaces that were created based on the maximum intensity projections. The FluoSpheres surfaces were enlarged using the radius-scaling function to improve visual perception. Finally, the images were processed using Adobe Photoshop CS version 5 (Adobe Systems, Mountain View, CA, USA).

### 4.14. Quantitative Image Analysis

The image stacks were analyzed using Imaris. Based on the maximum intensity projections, CD11b^+^ cells were processed via three-dimensional surface rendering of the APC channel. CD11b^+^ cell surfaces were quantified automatically. The inspection of the results was performed each time manually. Images of at least two regions proximate to the trachea from each whole-mount conducting airway specimen were acquired [48].

### 4.15. Statistical Analysis

The data are presented as the scattered dot plots with the median and i.q.r. for at least four mice per group. The differences between two groups were analyzed with the Mann–Whitney test using GraphPad Prism software (GraphPad Software, San Diego, CA, USA). A *p*-value less than 0.05 was considered statistically significant.

## Figures and Tables

**Figure 1 ijms-23-08234-f001:**
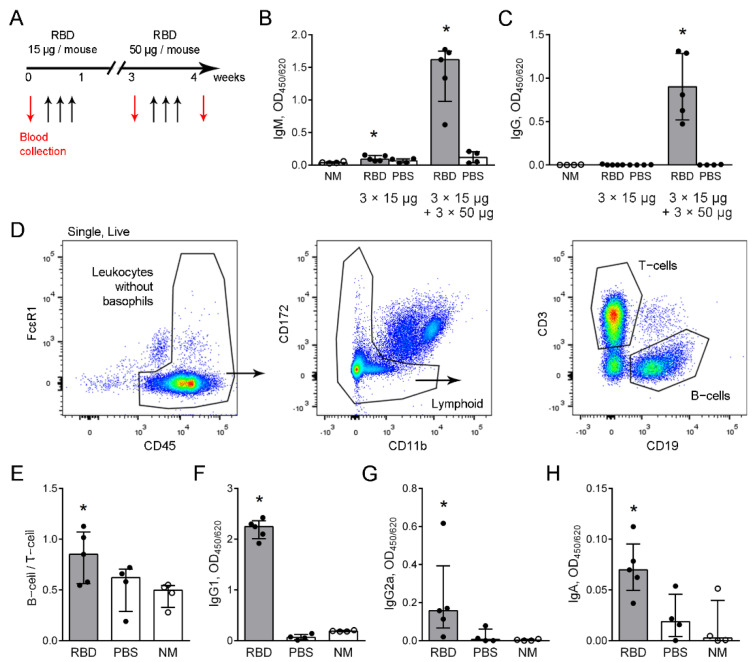
RBD–specific antibody induction. (**A**) The scheme of i.p. injections of RBD to mice (black arrows) and the time points of blood collection (red arrows). (**B**,**C**) The levels of IgM (**B**), serum dilution 1:100, and IgG (**C**), serum dilution 1:1000, two weeks after the last 15 µg RBD injection (3 × 15 µg) and a week after the last 50 µg RBD injection (3 × 15 µg + 3 × 50 µg). (**D**) The strategy of T− and B−cells detection in peripheral blood of the mice. (**E**) The ratio of B−cells to T−cells a week after the last RBD injection. (**F**–**H**) The levels of periphery blood IgG1 (**F**), IgG2a (**G**) serum dilution 1:1000, and IgA (**H**), serum dilution 1:100, in mice that were injected with RBD (gray bars, black circles) or in the control mice that were injected with PBS (open bars, black circles), and in the intact mice (NM, open bars, open circles) a week after the last RBD injection. The representative data are shown (*n* ≥ 4 mice per group). A significant difference between the indicated group and the intact mice was detected using Mann–Whitney test *: *p* ≤ 0.05.

**Figure 2 ijms-23-08234-f002:**
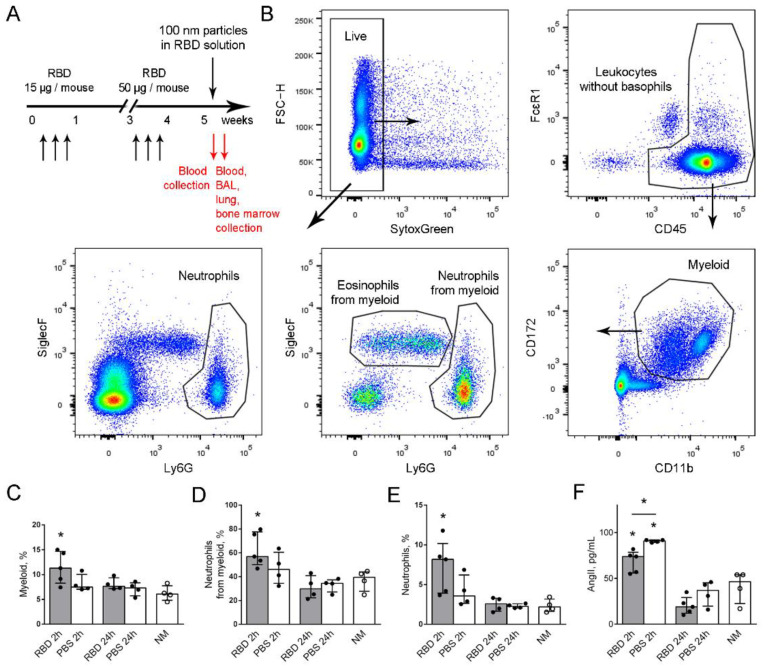
Neutrophil–mediated response to the application of 100 nm particles in the RBD solution. (**A**) The scheme of the RBD i.p. injections and o.ph. application of 100 nm particles in RBD solution to mice (black arrows). Two and 24 h after the o.ph application, blood was collected; 24 h after the application, BAL, lung and bone marrow were collected (red arrows). (**B**) The strategy of the neutrophil identification among blood cells or among myeloid cells. (**C**–**E**) The percentages of myeloid cells (**C**), neutrophils from myeloid cells (**D**) and neutrophils from live blood cells (**E**) 2 and 24 h after the o.ph. application of 100 nm particles in the RBD solution to the RBD—(gray bars, black circles) or PBS–injected mice (open bars, black circles); the intact mice NM (open bars, open circles). (**F**) The level of angiotensin II in the peripheral blood sera of mice injected with RBD (gray bars, black circles) or PBS (open bars, black circles) 2 and 24 h after the o.ph. application of 100 nm particles in the RBD solution, and the intact mice—NM (open bars, open circles). The data are shown as medians and interquartile range (i.q.r.) for *n* ≥ 4 mice per group. A significant difference between the indicated group and the intact mice was detected using the Mann–Whitney test *: *p* ≤ 0.05.

**Figure 3 ijms-23-08234-f003:**
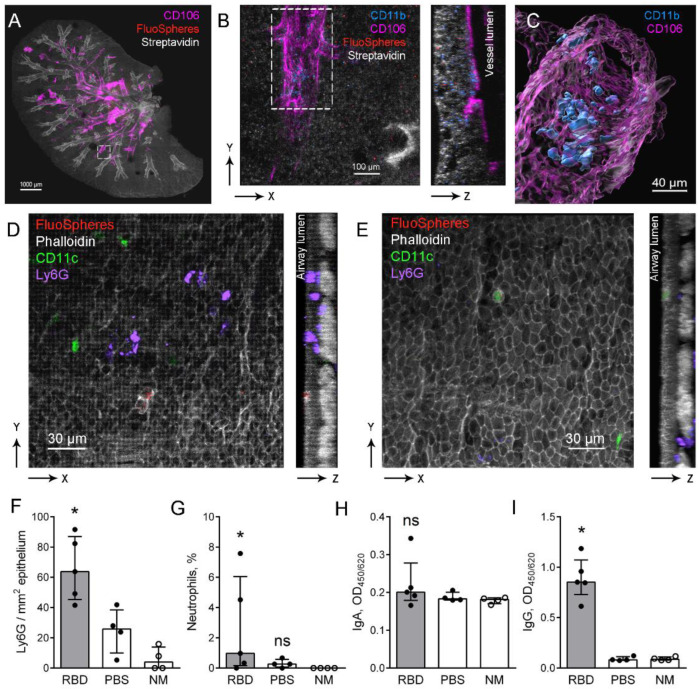
Identification and quantitative analysis of neutrophils in the airways and the lung vessels. (**A**) The representative three-dimensional image of the lung of mouse that received i.p. RBD injections and the o.ph. application of 100 nm particles in the RBD solution. The large vessels (CD106, magenta), the airways (Streptavidin, grayscale), and 100 nm particles agglomerates (FluoSpheres, red) are represented via volume rendering. Scale bar 1000 µm. (**B**) The higher magnification of the region boxed in (**A**) demonstrating the vessel (magenta) and CD11b^+^ cells (light blue) as the frontal (left image) and the lateral (right image) projections. Scale bar 100 µm. (**C**) The image of the region boxed in (**B**) showing the vessel and CD11b^+^ cells via surface rendering. Scale bar 40 µm. (**D**,**E**) Representative three-dimensional images of the regions of conducting airway mucosa of the mouse with induced RBD-specific antibodies (**D**) and of the PBS-injected mouse (**E**) 24 h after the application of 100 nm particles in the RBD solution. 100 nm particles (FluoSpheres, red), epithelial and smooth muscle cells (phalloidin, grayscale), neutrophils (Ly6G, light blue) and CD11c^+^ cells (green) are represented as the frontal (upper images) and the lateral (lower images) projections. Scale bar 30 µm. (**F**,**G**) The quantitative analysis of neutrophils in the airway mucosa (**F**) or BAL (**G**) of mice that were injected with RBD (gray bars, black circles) or with PBS (open bars, black circles) 24 h after the application of 100 nm particles in the RBD solution, and in intact mice (NM, open bars, open circles). (**H**,**I**) The levels of IgA (**H**) and IgG (**I**) in the BAL fluids of mice that were injected with RBD (gray bars, black circles) or with PBS (open bars, black circles) 24 h after the application of 100 nm particles in RBD solution, and in the intact mice (open bars, open circles). The data are shown as medians and i.q.r. for *n* ≥ 4 mice per group. A significant difference between the indicated group and the intact mice was detected using the Mann–Whitney test *: *p* ≤ 0.05.

**Figure 4 ijms-23-08234-f004:**
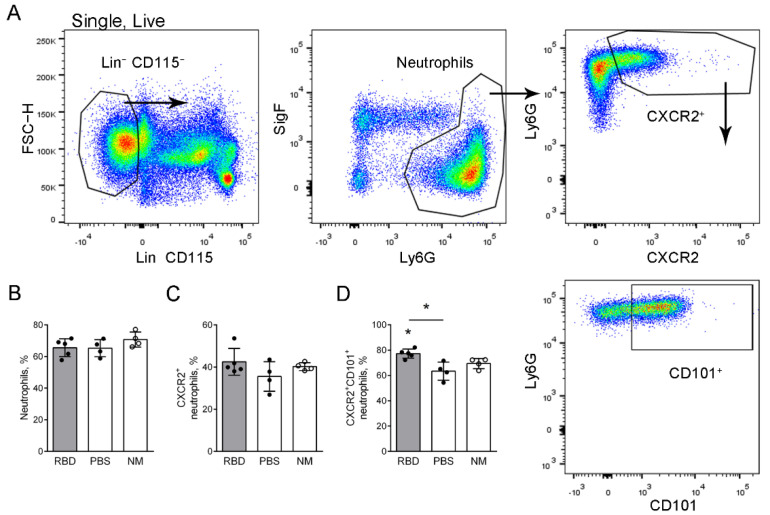
RBD–specific antibody induction effect on the bone marrow neutrophil activation and maturation. (**A**) The strategy of CXCR2^+^ and CD101^+^ neutrophil detection in the bone marrow of the mice. (**B**–**D**) The percentages of neutrophils (**B**), CXCR2^+^ neutrophils (**C**), and CXCR2^+^CD101^+^ neutrophils (**D**) in the bone marrow of mice that were injected with RBD (gray bars, black circles), PBS (open bars, black circles) 24 h after the application of 100 nm particles in the RBD solution, and intact mice (open bars, open circles). The data are shown as medians and i.q.r. for *n* ≥ 4 mice per group. A significant difference between the indicated group and the intact mice or between the indicated groups was detected using the Mann–Whitney test *: *p* ≤ 0.05.

**Figure 5 ijms-23-08234-f005:**
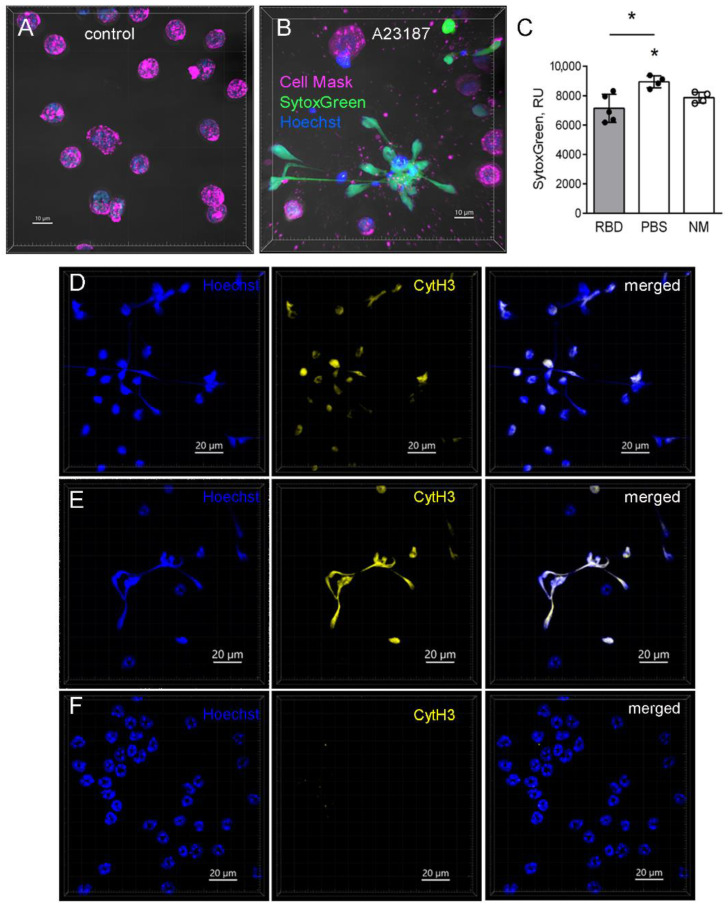
NET formation by bone marrow neutrophils from mice with RBD–specific antibodies. (**A**,**B**) The representative images of neutrophils yielded from the bone marrow of the PBS–injected mouse 3 h after the incubation in the presence (**B**) or without (**A**) of 5 µm of A23187. The neutrophils were stained with Cell mask (magenta), SytoxGreen (green), and Hoechst (blue). Scale bar 10 µm. (**C**) The quantitative analysis of the extracellular nucleotide levels 3 h after the incubation with A23187 of neutrophils from the bone marrow of mice injected with RBD (gray bars, black circles), PBS (open bars, black circles) or intact mice (open bars, open circles). Data are shown as medians and i.q.r. for *n* ≥ 4 mice per group. A significant difference between the indicated group and the intact mice or between the indicated groups was detected using the Mann–Whitney test *: *p* ≤ 0.05. (**D**–**F**). The representative images of nuclei (Hoechst, left images), citrullinated histone H3 (CytH3, middle images), and co-localization (merged, right images) are presented for the A23187–treated (**D**,**E**) and untreated neutrophils (**F**), yielded from RBD–injected (**D**) or PBS-injected (**E**,**F**) mouse 24 h after the application of 100 nm particles in the RBD solution. Scale bar 20 µm.

## Data Availability

The original contributions presented in the study are included in the article/Appendix A, further inquiries can be directed to the corresponding author.

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
