# Peer review of "Short-Term Effect of SARS-CoV-2 Spike Protein Receptor-Binding Domain-Specific Antibody Induction on Neutrophil-Mediated Immune Response in Mice"

_ijms, 2022, doi:10.3390/ijms23158234_

Round 1

Reviewer 1 Report

Bolkhovitina et al. provides an interesting experiment in order to observe and characterize the different behavior of neutrophils in a COVID19 vaccineted  mouse model , compared to controls, after exposure to spike protein receptor binding domain.

The authors demonstrated that , despite a higher neutrophil recruitment, the immunized mice are more mature and are less reactive to neutrophil extracellular traps (NET) after exposure to covid spike protein when compared to the control mice not immunized. This is of relevance, because could explain  why there is less symptomatology in vaccinated subjects.

Nevertheless, the authors might add a comment/evaluation on the implication that this result might have in a translational sense on humans.

In the abstract I advise to add the extended versione for NET (that is missing)

Author Response

Dear Reviewer,

Thank you very much for the comprehensive review.

We improved grammar and clarity of the manuscript, using the assistance of the English language expert.

According to your recommendation, we have added possible implications of our results in clinical study to the discussion section.

We have also inserted the NETs abbreviation expansion to the abstract.

Thank you again for reading and evaluating our manuscript.

Reviewer 2 Report

In this study, the authors, using a mouse model mimicking SARS-CoV-2 virus particle inhalation, investigated a bone marrow neutrophil phenotype and activity alterations in the presence or absence of receptor-binding domain (RBD)-specific antibodies. Mice were immunized with RBD and a week after a strong antibody response establishment received 100 nm particles in the RBD solution. Control mice received injections of a phosphate buffer instead of RBD.

They showed that application of 100 nm particles in the RBD solution induces elevated neutrophil recruitment to the blood of RBD-immunized mice rather than in control mice. Elevated neutrophil numbers were detected in the airways of RBD-immunized, but not control mice, after the application of 100 nm particle in the RBD solution.

The analysis of bone marrow neutrophils of mice with induced RBD-specific antibodies demonstrated the increased population of CXCR2+CD101+ neutrophils. These neutrophils did not demonstrate enhanced ability of NETs formation compared to the neutrophils from control mice.

They concluded that induction of RBD-specific antibodies stimulates activation of mature neutrophils that react to RBD-coated particles without triggering excessive inflammation.

The study is of interest with findings supporting an interplay between specific activation of adaptive and neutrophil-mediated innate immune response. also useful to explain both the protective and side effects of vaccination.

I have only a comment on this interesting study. It is reported that the intricate immune response mechanism in SARS-CoV-2 virus disease may induce also autoantibody production. The authors should recall recent  literature data demonstrating that severe interstitial lung disease in COVID-patients are significantlyassocited to the development of antinuclear antibodies (ANA), in particular ANA exhibiting the "nucleolar" pattern, as demonstrated and reported in recent studies (COVID-19 and Immunological Dysregulation: Can Autoantibodies be Useful? Clin Transl Sci. 2021 Mar;14(2):502-508; Antinuclear antibodies in COVID 19. Clin Transl Sci. 2021 Sep;14(5):1627-1628).

The authors should also discuss the reported role of SARS-CoV-2 virus as potential trigger of autoantibodies and autoimmunity as demonstrated by the occurrence of other autoantibodies (Coronavirus disease associated immune thrombocytopenia: Causation or correlation? J Microbiol Immunol Infect. 2021 Jun;54(3):531-533; Autoantibodies and autoimmune disorders in SARS-CoV-2 infection: pathogenicity and immune regulation. Environ Sci Pollut Res Int. 2022 Jun 3:1–16. )

Author Response

Dear Reviewer,

Thank you very much for the detailed review.

In accordance with your recommendation, using the assistance of the English language expert, we improved grammar and clarity of the manuscript.

Thank you very much for the very insightful comment regarding the role of autoantibodies in COVID-19-associated complications. With great interest, we have read the indicated papers and cited them in the discussion section. We also added some thoughts about ANA, ANCA and NET formation interplay in COVID-19- and vaccination-associated complications.

Thank you for the useful feedback.